

# The effects of assimilating a sub-grid scale sea ice thickness distribution in a new Arctic sea ice data assimilation system

Nicholas Williams[1], Nicholas Byrne[2], Daniel Feltham[1], Peter Jan Van Leeuwen[23], Ross Bannister[2], David Schroeder[1], Andrew Ridout[4], and Lars Nerger[5]

[1]Centre for Polar Observation and Modelling, Department of Meteorology, University of Reading, Earley Gate, PO Box 243, Reading, RG6 6BB, UK
[2]Department of Meteorology, University of Reading, Earley Gate, PO Box 243, Reading, RG6 6BB, UK
[3]Department of Atmospheric Science, Colorado State University, Fort Collins, CO 80523, United States
[4]Centre for Polar Observation and Modelling, Department of Earth Sciences, University College London, London, UK
[5]Alfred Wegener Institute, Helmholtz Center for Polar and Marine Research, Bremerhaven, Germany

**Correspondence:** Nicholas Williams (n.p.williams@pgr.reading.ac.uk)

**Abstract.** In the past decade groundbreaking new satellite observations of the Arctic sea ice cover have been made, allowing researchers to understand the state of the Arctic sea ice system in greater detail than before. The derived estimates of sea ice thickness are useful but limited in time and space. In this study the first results of a new sea ice data assimilation system are presented. Observations assimilated (in various combinations) are monthly mean sea ice thickness and monthly mean sea ice thickness distribution from Cryosat-2, and NASA daily Bootstrap sea ice concentration. This system couples the Centre for Polar Observation and Modelling's (CPOM) version of the Los Alamos Sea Ice Model (CICE) to the Localised Ensemble Transform Kalman Filter (LETKF) from the Parallel Data Assimilation Framework (PDAF) library. The impact of assimilating a sub-grid scale sea ice thickness distribution is of particular novelty. The sub-grid scale sea ice thickness distribution is a fundamental component of sea ice models, playing a vital role in the dynamical and thermodynamical processes, yet very little is known of its true state in the Arctic.

This study finds that assimilating Cryosat-2 products for the mean thickness and the sub-grid scale thickness distribution can have significant consequences on the modelled distribution of the ice thickness across the Arctic and particularly in regions of thick multi-year ice. The assimilation of sea ice concentration, mean sea ice thickness and sub-grid scale sea ice thickness distribution together performed best when compared to a subset of Cryosat-2 observations held back for validation. Regional model biases are reduced: the thickness of the thickest ice in the Canadian Archipelago is decreased, but the thickness of the ice in the Central Arctic is increased. When comparing the assimilation of mean thickness with the assimilation of sub-grid scale thickness distribution, it is found that the latter leads to a significant change in the volume of ice in each category. Estimates of the thickest ice improve significantly with the assimilation of sub-grid scale thickness distribution alongside mean thickness.



# 1    Introduction

Arctic sea ice is an important indicator of climate change and plays a key role in the global surface energy balance. Depending on its thickness, its surface reflects 50-75 percent of incoming solar radiation and it impedes exchanges of heat, moisture and momentum between the atmosphere and the ocean. Sea-ice freeze-up and melt also effects deep water formation and the thermohaline circulation. Since the beginning of the industrial era (1850+), surface temperatures have risen by approximately 1 degree Celsius on average worldwide, but Arctic amplification has seen temperatures in the Arctic rise at roughly twice this (Serreze and Barry, 2011). The primary (but not solely responsible) mechanism driving Arctic amplification is a declining sea ice cover observed over the Satellite era (1979+) (Dai et al., 2019). Observations of the sea ice state are sparse, and only observations of sea ice concentration and motion are available over the whole extent of the satellite period. Sea ice freeboard – and consequently thickness observations – have only been attainable in the past two decades, and provide valuable information on the sea ice state.

In order to better understand the changes in the Arctic sea ice that have occurred over the satellite era, data assimilation (DA) can be used to fill in the spatial and temporal gaps in these observations. Sea ice in the Arctic is not spatially or temporally uniform but varies with mixtures of thinner first-year ice and thicker multi-year ice. Describing the variation and evolution of the sea ice thickness in space and time is the goal of sea-ice modelling, but presently the sea ice state in the Arctic is poorly known. Recently observations of the ice thickness distribution in the Arctic have been derived from Cryosat-2 (CS2) measurements (Schröder et al., 2019) and in this paper we provide new insights by using these new data to produce a partial history (2012-2015) of the sea ice thickness distribution and its transformation during this period of minimum sea ice extent. The present study is intended to pave the way for a 40-year reanalysis of Arctic sea ice using the system described.

The earliest efforts to combine data assimilation techniques with large scale numerical sea ice models began in the early 1990s with work by Thomas and Rothrock (1993) on assimilating sea ice concentration data using Kalman smoother techniques. As sea ice concentrations were the only available sea ice observations at this time most of the work on sea ice DA only used sea ice concentration observations, including in the pan-Arctic Ice Ocean Modeling and Assimilation System (PIOMAS) reanalysis (Zhang and Rothrock, 2003). Sea ice concentration is assimilated in PIOMAS using a relatively simple optimal interpolation technique which aims to move the model variables closer to their observed counterparts through a weighting factor. Sea surface temperatures are assimilated in PIOMAS through optimal interpolation. PIOMAS has been extensively validated through satellite and in situ (submarine, mooring) observations and the uncertainty for October sea ice volume estimations is believed to be $1.35 \times 10^3 \mathrm{km}^3$ (Schweiger et al., 2011). Sea ice motion observations were also an early focus of sea ice data assimilation (Meier et al., 2000), available from ocean buoys and remotely sensed data. There were also some early efforts at assimilating synthetic ice thickness data (Lisæter et al., 2007) which showed how a sea ice model may be impacted by sea ice thickness assimilation.





Within the past decade a growing number of ocean-sea ice DA systems and reanalyses have been produced and are in operation, including the UK Met Office's FOAM/GloSea5 (Blockley et al., 2014), NERSC's TOPAZ4 (Sakov et al., 2012), ECMWF's Ocean ReAnalysis Pilot 5 (ORAP5) (Zuo et al., 2017) and the MERRA Ocean product (Rienecker et al., 2011). Many of these use 3D-Var or Optimal Interpolation (OI) and assimilate only sea ice concentration. TOPAZ4 uses an Ensemble Kalman Filter (EnKF) for sea ice concentration and has tested the assimilation of CS2-SMOS sea ice thickness (Ricker et al., 2017). The Met Office has tested sea ice thickness assimilation in its FOAM system with OI assimilation of CS2 monthly mean ice thickness (Blockley and Peterson, 2018), and recently using 3D-Var assimilation of its daily sea ice thickness data (Fiedler et al., 2022). They have also recently tested the model with assimilation of the combined Cryosat-2-Soil Moisture and Ocean Salinity (CS2-SMOS) product (Mignac et al., 2022). A comparison of fourteen ocean-sea ice reanalyses has found that the spatial pattern of ice volume varies widely between products, with no reanalysis standing out as clearly superior when compared to altimetry estimates, ice thickness from sea ice models without assimilation of sea ice concentration is not worse than that from systems constrained with sea ice observations. Many of these studies also use PIOMAS to calibrate ice thickness simulations, but PIOMAS does not assimilate sea ice thickness (Chevallier et al., 2017). A recent sea ice DA experiment is that of Fritzner et al. (2019), which assimilated sea ice concentration, ice thickness from both CS2 and SMOS, and a recently developed snow depth observational product from Rostosky et al. (2018) using CICE and an EnKF. At the Geophysical Fluid Dynamics Laboratory (GFDL), researchers have further developed their seasonal prediction system to include the assimilation of sea ice concentration using the Ensemble Adjustment Kalman Filter (EAKF), which significantly improves seasonal predictions of sea ice extent (Zhang et al., 2021). The assimilation of Ocean and Sea Ice Satellite Application Facility (OSISAF) sea ice concentration, OSISAF sea ice drift, CS2-SMOS sea ice thickness and sea surface temperature has also been studied in a global climate model at the Alfred Wegener Institute (AWI-CM v1.1). This uses a one-category thickness distribution, zero-layer thermodynamics scheme and Elastic-Viscous-Plastic (EVP) sea ice rheology and assimilates using a Local Error Subspace Transform Kalman Filter (LESTKF). This has been found to perform well against independently available in situ data (Mu et al., 2020).

In the current study we use a sophisticated sea-ice model, the Centre for Polar Observation and Modelling (CPOM) version of CICE in a 100 ensemble member Localised Ensemble Transform Kalman Filter to assimilate monthly mean ice thickness, and – for the first time – sub-grid scale ice thickness distribution observations from CS2. We validate the effectiveness of this assimilation by randomly choosing 75% of CS2 observations for assimilation, with the remaining for validation. The sub-grid ice thickness distribution is a vital component of any sea ice model and yet our knowledge of its true state in the Arctic is poor. In Sect. 2 we outline the sea ice model and DA framework used to produce the sea ice DA system. In Sect. 3 we discuss the observations of sea ice we use for assimilation and evaluation. In Sect. 4 we conduct studies to find ideal assimilation settings and then use these to produce a four year reproduction of the Arctic sea ice state between 2012 and 2015. Furthermore, we compare and validate it against randomly selected CS2 observations that are not assimilated, and the PIOMAS sea ice reanalysis. We look at how the modelled sea ice extent, thickness and volume differ when only assimilating sea ice concentration and when additionally assimilating mean thickness and sub-grid scale thickness distribution to try and understand how the assimilation





effects the distribution of the ice thickness in the Arctic. In Sect. 5 we discuss the consequences of these results for the Arctic
sea ice and the CPOM-CICE model, and any potential drawbacks of the sea ice data assimilation system we use. We conclude
by discussing the key outcomes from this paper and our future research direction.

## 2  Method

### 2.1  Sea ice model

We use the Centre for Polar Observation and Modelling (CPOM) version of the Los Alamos Natural Laboratory Sea Ice Model
(CICE) v5.1.2 (Hunke et al., 2015) with five ice thickness categories. CICE is a dynamic and thermodynamic model of sea
ice designed to function as the sea ice component of a fully coupled global climate model. We use the incremental remapping
scheme of Lipscomb and Hunke (2004) to solve the horizontal transport equation, with the internal stress tensor in the sea
ice momentum balance determined using the elastic-plastic-anisotropic rheology of Tsamados et al. (2013). The model also
includes a parameterization of form drag (Tsamados et al., 2014). For the thermodynamic model, we use the 1-dimensional
vertical Bitz and Lipscomb model (Bitz and Lipscomb, 1999) which solves heat flux balance equations for the ice and snow
(if it exists) in each category and accounts for the effects of brine pocket melting and freezing. We use the Delta-Eddington
approach for computing the ice shortwave albedo and incoming shortwave radiation, where the natural optical properties for
snow, sea ice and melt ponds are used to define their scattering and absorption characteristics (Briegleb and Light, 2007). This
Delta-Eddington approach is used in conjunction with the topographic melt pond scheme of Flocco et al. (2010). The bubbly
brine thermal conductivity parameterization is also used, which increases the conductivity of colder sea ice (Pringle et al.,
2007). The transport of ice in thickness space due to the thermodynamic changes in the sea ice uses the remapping scheme of
Lipscomb (2001). The mechanical redistribution of the ice thickness is formulated by the scheme of Lipscomb et al. (2007)
with ice strength defined as in Rothrock (1975).

Although CICE was designed for use in a global climate model, in this research we use it in stand-alone mode, uncoupled
to an atmospheric or ocean model. The model has been used in this way to produce realistic estimates of the sea ice state, e.g.
Schröder et al. (2019), and its computational efficiency facilitates physical and technical model development. CICE contains a
thermodynamic slab mixed layer ocean model with a prognostic ocean temperature. This model is initialised with ocean tem-
perature and salinity (3m depth) from a 1993-2010 climatology based on an ocean reanalysis (Ferry et al., 2011). The mixed
layer salinity is prescribed from the climatology and the prognostic temperature is restored to the monthly climatology with a
20 day timescale to account for heat advection in the ocean. The ocean currents (also at 3m depth) are also taken from the same
reanalysis. The atmospheric forcing data used are NCEP-2 (Kanamitsu et al., 2002) comprising daily downward shortwave and
longwave radiation fluxes and 6-hourly 2m temperature and humidity and 10m wind velocity. These atmospheric forcing fields
are perturbed to generate ensemble spread as described in Sect. 2.3. We also take monthly mean precipitation from the same
reanalysis, which is not perturbed. We use an ORCA 1 degree (a roughly 40km by 40km grid size) tripolar grid, covering the
whole Arctic region. A time step of 1 hour is used.





## 2.2 The Ensemble Kalman Filter

The parallel data assimilation framework (Nerger and Hiller, 2013), known hereinafter as PDAF, is a software environment designed to provide ensemble based DA algorithms that can be implemented within existing numerical models with only minimal changes to the model code. We are using PDAF V1.16, which improved observation handling over previous versions of PDAF through the use of a modular implementation. PDAF currently contains the ability to implement many types of EnKFs, such as the LETKF, the LESTKF and the stochastic EnKF. In our sea ice DA system we have coupled CICE to PDAF, and are using the LETKF (Hunt et al., 2007). The basic Kalman filter is a sequential DA method, which solves for the mean and covariance of the posterior's probability density function (PDF) assuming it is Gaussian (instead of solving for the mode of the posterior PDF, as in variational methods). The equations for the Kalman gain and the mean and covariance analysis updates are

$$\mathbf{K}_n = \mathbf{P}_n^{\mathrm{f}} \mathbf{H}^{\mathrm{T}} (\mathbf{H} \mathbf{P}_n^{\mathrm{f}} \mathbf{H}^{\mathrm{T}} + \mathbf{R})^{-1}$$

$$\mathbf{x}_n^{\mathrm{a}} = \mathbf{x}_n^{\mathrm{f}} + \mathbf{K}_n (\mathbf{y}_n - \mathbf{H} \mathbf{x}_n^{\mathrm{f}})$$

$$\mathbf{P}_n^{\mathrm{a}} = (\mathbf{I} - \mathbf{K}_n \mathbf{H}) \mathbf{P}_n^{\mathrm{f}},$$

where $\mathbf{K}_n$ is the Kalman gain at time $n$, $\mathbf{P}_n^{\mathrm{f}}$ is the forecast error covariance, $\mathbf{H}$ is the observation operator, $\mathbf{R}$ is the observation error covariance matrix, $\mathbf{x}_n^{\mathrm{a}}$ is the analysis model state, $\mathbf{x}_n^{\mathrm{f}}$ is the forecast model state, $\mathbf{y}_n$ is the observation vector and $\mathbf{P}_n^{\mathrm{a}}$ is the analysis error covariance. The errors are assumed to have Gaussian statistics with zero mean. In the Kalman filter, it is necessary to propagate the error covariance matrices using tangent-linear and adjoint models. This is not feasible in practice because of the large size of the covariance matrix. To avoid this, an ensemble of states – which sample from the uncertainty at time $n$ – are used to approximate $\mathbf{P}_n^{\mathrm{f}}$. Then, post assimilation, each member is evolved forward using the model, thus creating an EnKF. The mean and covariance are then sampled at each assimilated time step.

We have chosen to use an EnKF, rather than variational methods, because it avoids needing tangent linear and adjoint $\mathbf{H}$ matrix, and avoids the need to parametrise a $\mathbf{P}^{\mathrm{f}}$-matrix. Each of these would be time consuming and difficult. Using the EnKF system also means that we can assimilate observations sequentially, rather than assimilating multiple observations in a window, and it automatically yields (implicitly) a flow-dependent $\mathbf{P}^{\mathrm{f}}$-matrix from the ensemble. This means though that EnKFs are susceptible to ensemble collapse if the ensemble spread is not sufficient (Houtekamer and Mitchell, 1998). If ensemble collapse occurs then the assimilation becomes ineffective because the EnKF will measure the forecast error to be anomalously small. As the EnKF is sensitive to ensemble size and undersampling we can use both inflation (through use of a forgetting factor), enhanced atmospheric forcing, and localisation to mitigate this. The LETKF implemented in PDAF makes use of a forgetting





factor $\rho$ (Pham et al., 1998) which is applied in a computationally more efficient way than an inflation factor $r$. The two are related through $\rho = r^{-2}$.


To avoid spurious correlations due to the relatively small ensemble size we use localisation, in which only observations within a pre-specified localisation radius of each grid point are used to update the variables at that grid point. In CICE-PDAF we make use of a Gaspari and Cohn function (Gaspari and Cohn, 1999) as a weighting function for the observation error covariance, which is scaled such that the effects of the observation smoothly reach zero at and beyond a defined localisation radius away

from the observation. We use a range of $\rho$ and $r_l$ values in our experiments, which are presented in Sect. 4.

### 2.3 Generating Ensemble Spread

CICE is a forced dissipative model, and thus if we were to run it with $n$ members under the same set of forcing parameters, we would not find enough ensemble spread necessary to produce a working DA system. Ensemble spread is vital for any ensemble based DA to function correctly as it is used to capture the uncertainty in the model state. In CICE-PDAF we apply an

EOF-based perturbation method (**?**) to the 6-hourly NCEP-2 reanalysis atmospheric forcing fields (see Sect.2.1). By applying the perturbations indirectly to the forcing in this way we preserve the dynamic and thermodynamic consistencies within each forcing field. For each month we perform a multivariate analysis on each of the six atmospheric forcing fields. We then choose the number of EOF modes that represent the majority (95%) of the variability in each of the forcing fields. We then add perturbations to the original forcing fields for each ensemble member using


$$\sum_{i=1}^{j} \alpha N_i \sigma_i EOF_i$$

where $i$ is the EOF mode, $j$ the total number of EOFs chosen to represent the variance in the field, $N_i$ are the random numbers

chosen from a normal distribution with zero mean and unit variance, $\sigma_i$ are the climatological standard deviations of that forcing field in a given month, and $\alpha$ is a multiplicative factor that we can use to amplify the perturbations on the atmospheric forcing to induce additional ensemble spread. We have used $\alpha = 1.5$. For each month in a particular year, the same random numbers are chosen for each field for the resulting perturbations to maintain consistency between each forcing field as they were in the original forcing.

### 2.4 Post analysis step processing

Due to the Gaussian assumption of the EnKF update, many of the CICE model state variables can move outside of their physical bounds, or cause a lack of physical consistency between state variables. In order to rectify this, the analysis ensemble states are verified by a series of post analysis checks done inside PDAF, before they are sent back to CICE. This proceeds as follows: in the situation where the assimilation creates ice in a grid cell where there was none previously, we must prescribe the snow





enthalpy, ice enthalpy, salinity and surface temperatures. For model consistency this is done using the same thermodynamic layer model used in CICE, using a linear temperature profile with a predefined salinity profile and melting temperature. Once this is done we check the bounds of a number of CICE state variables for all situations (whether ice cover is created, destroyed or just modified): namely salinities, enthalpies, melt pond parameters and surface temperatures. We then reset negative ice areas, ice volumes and snow volumes to zero. If total ice fractional areas exceed unity, these are normalised to unity, such

that the fractions of ice in each thickness category determined by the EnKF are preserved; ice and snow volumes are reduced accordingly in line with this normalisation. Additionally it was also found that anomalously large sea ice thickness can occur on the ice edge where the EnKF reduces the ice concentration to a very small amount (lower than $10^{-5}$) but ice volume is not reduced in the same way. These lead to ice thickness spikes, which can cause CICE to crash. In these situations ice in these grid cells is removed. If ice within a thickness category in a grid cell is totally destroyed then the CICE state variables required to

be consistent with this are also reset. The exact variables which may be modified in this post-processing are specified in Table 1. CICE itself also contains its own routines that check for consistency and correct physical properties within the state vector.

**Table 1** A list of CICE state variables altered during post-processing.

| state variable | physical meaning | checks |
|---|---|---|
| qice(001-007) | sea ice enthalpy in the vertical ice layers (1-7) | within physical bounds and physically consistent |
| sice(001-007) | sea ice salinity in the vertical ice layers (1-7) | within physical bounds and physically consistent |
| qsno001 | snow enthalpy | within physical bounds and physically consistent |
| aicen | fractional sea ice area | within physical bounds, normalised if total concentration above 1. |
| vicen | sea ice volume | within physical bounds, normalised if total concentration above 1 |
| vsnon | snow volume | within physical bounds, normalised if total concentration above 1 |
| apnd | melt pond area | within physical bounds |
| hpnd | melt pond depth | within physical bounds |
| ipnd | melt pond refrozen lid thickness | within physical bounds |
| Tsfcn | snow/ice surface temperature | must be $> -70°C$ |

## 3 Observation and Evaluation Data

### 3.1 Sea Ice Concentration

Sea ice concentration has been observable by passive microwave satellites since the early 1970s. Modern datasets of sea ice concentration are derived from brightness temperature observations from satellite instruments such as the Scanning Multichan-

nel Microwave Radiometers (SMMR) which were launched in 1978, the Special Sensor Microwave/Imagers (SSM/I) launched between 1987-99 and the Special Sensor Microwave Imager Sounder (SSMIS) launched between 2003-14. SSM/I data sets are extremely useful as they have provided a continuous time series of sea ice information since their launch – the most useful for



us being sea ice concentration. The most widely used algorithms for obtaining sea ice concentration are the NASA Team algorithm (Markus and Cavalieri, 2000) and the NASA Bootstrap (Comiso, 2017). Although based upon the same key principle, the

algorithms differ in a number of key ways: different choices in both frequency channels and polarizations, choice of reference brightness temperatures, and their sensitivity to surface temperature and surface characteristics. The NASA Team algorithm looks at contributions from three different surface types: MYI, FYI and ice-free ocean (Cavalieri, 1991), and uses the 19V, 19H and 37V channels from the SSM/I, SSMR and SSMIS sensors with a resolution of 25km. The radiances from these channels are then used to calculate a polarization ratio and spectral gradient ratio from which sea ice concentration can be retrieved.

The Bootstrap algorithm also uses the SSM/I, SSMR and SSMIS but instead takes advantage of the correlated distributions of the brightness temperatures over the Arctic, specifically over the 37GHz channels. Unlike the Team algorithm however, it only derives a single sea ice concentration rather than separate FYI and MYI concentrations. In comparisons between The Team and Bootstrap algorithms (Comiso et al., 1997) it was found that both generally show similar ice-edge positions for the Arctic, though the Team algorithm in general derives smaller ice concentrations, especially during January and around the inner ice

pack. High temperature and emissivity fluctuations in the marginal ice pack can also produce discrepancies (Comiso et al., 1997), which can lead to large differences in sea ice concentration calculations. The NASA team algorithm is also better at handling the brightness temperature fluctuations. Another key difference lies in their handling of melt ponds, which appear as open water in the satellite data (Comiso et al., 1997). The Bootstrap algorithm tries to offset this bias by synthetically increasing estimates of sea ice concentrations much more than the Team algorithm does (Bunzel et al., 2018).

For the following DA experiments we choose to assimilate Bootstrap sea ice concentration because it attempts to account for the interferences in the radiometers caused by the melt ponds. Sea ice concentration retrieval errors are difficult to quantify but the overall retrieval accuracy is estimated to be between 5 and 10%. However, there are significant issues with measuring newly formed sea ice (Comiso et al., 1992) because of the changes in the emissivity of sea ice as it grows, and with melt ponds (Comiso and Kwok, 1996) as these appear as open water in satellite data. This causes particular problems over the summer

melting period as melt ponds can dominate the surface of the Arctic sea ice cover. The Bootstrap algorithm tries to offset this by synthetically increasing its estimates of sea ice concentrations. There are also issues with the ice concentration retrieval arising from atmospheric conditions, thin ice and snow melt (Ivanova et al., 2015), in these grid cells error estimates could be as high as 30% (Comiso, 2017). Grid cells In CICE-PDAF we use sea ice concentration error standard deviations that depend on the time and concentration. For sea ice concentration above 0.8, we use an error standard deviation of 0.2 for May-September

(inclusive) and 0.1 outside of these months. Otherwise we use an error standard deviation of 0.15. The model version of the observation is found easily just by summing the individual ice concentrations in each of the thickness categories, and the ice concentrations in each category are updated in the EnKF through the correlations between themselves and the total sea ice concentration.

## 3.2  Sea Ice Thickness

The sea ice thickness observations we assimilate are from the CPOM derived monthly mean CS2 sea ice thickness product (Laxon et al., 2013). Cryosat-2 is a European Space Agency (ESA) mission whose primary aim is to observe trends in Earth's



continental and marine ice fields (Wingham et al., 2006). It uses a Synthetic Aperture Interferometric Radar Altimeter (SIRAL) to monitor up to 88 degrees North. This works by transmitting microwave pulses at regular intervals defined by a pulse repetition frequency towards the Earth and measuring the time taken by the pulse to reflect off the Earth's surface and return to the
satellite. CPOM uses such radar altimeter data from CS2 and processes them to produce Arctic sea ice thickness and volume datasets (Tilling et al., 2018).

There are two important processes in deriving sea ice thickness measurements from the raw observation data: the differentiation between measurements of surface elevation of the ocean and surface elevation of the sea ice, and the process which then
converts the calculated freeboard to thickness. Sea ice and the leads between ice floes can be differentiated in radar echoes by the shape of the echo waveform, and then a process known as retracking is used to determine the location on each waveform which represents the average surface elevation within the satellite footprint. CPOM uses a Gaussian-plus-exponential waveform fit to retrack echoes from leads (Giles et al., 2008) and a 70% leading edge threshold from the first peak to retrack floe echoes. Other retrackers have been used to produce freeboard and thickness data from CS2, which can yield markedly different
results (Ricker et al., 2014). It is assumed that the radar bursts reflect off the snow-ice interface as shown in (Beaven, 1995) though more recent research shows that this may not always be the case (Stroeve et al., 2020). The biggest source of uncertainty occurs when converting freeboard to sea ice thickness, which is found using

$$h_i = \frac{f_c \rho_w + h_s \rho_s}{\rho_w - \rho_i}.$$


where $h_i$ is sea ice thickness, $f_c$ is the sea ice freeboard, $\rho_w$ is the density of seawater, $h_s$ is snow depth, $\rho_s$ is snow density and $\rho_i$ is sea ice density, which is different depending on whether it is FYI or MYI. Thus to acquire measurements of sea ice thickness we require knowledge of the snow cover on the Arctic sea ice. Since such snow measurements are very limited, CPOM uses snow depth from a modified Warren climatology (Warren et al., 1999). This is a climatology of snow depth derived
from in situ data gathered from Soviet drifting stations on MYI from 1954-1991. This climatology may no longer be valid due to the large observed loss of Arctic sea ice in the past 2 decades - and in particular loss of MYI. For instance snow depth on FYI is found to be half of that found in the Warren climatology (Kurtz and Farrell, 2011), so the modified climatology used by CPOM halves snow depth on FYI. Use of the Warren climatology is necessary because snow depth satellite retrievals are not currently feasible, and there are still large problems to overcome in snow loading models, such as uncertainties from the mag-
nitude of precipitation (Serreze and Hurst, 2000), unknown snow compaction and difficulties in estimating snow loss caused by leads. Due to these uncertainties, CS2 ice thickness measurements of thinner ice, particularly below 0.5 metres, can have very large error estimates, because of uncertainties in the snow depth. In our assimilation we use a relative observation error dependent on the observed ice thickness measured, with sea ice thicker than 5 m using a 50% relative error, and thinner ice using a 25% relative error (Tilling et al., 2018). Ice thinner than 0.5 m is not assimilated due to the high uncertainty associated
with Cryosat-2 measurements of the thinnest ice. As there is a significant lack of sea ice thickness data available for validation, we choose to assimilate approximately 75% of the available monthly CS2 data in our assimilation and then use the remainder





for validation. This is done by picking a random number between 0 and 1 for each grid cell and if this number is below 0.75 data in that grid cell will be assimilated, otherwise it will be used for validation.

## 3.3 Sea Ice Thickness Distribution

The problem that sea ice models seek to solve is the evolution of the sea ice thickness distribution, $g$

$$\frac{\partial g}{\partial t} = -\nabla \cdot (g\mathbf{u}) - \frac{\partial}{\partial h}(fg) + \psi,$$

where $\mathbf{u}$ is the ice velocity, $f$ is the rate of thermodynamic ice growth and $\psi$ is the ridging distribution function. $g$ is solved in sea ice models by splitting the ice in each grid cell into thickness categories and replacing $g$ in the equation above with $a_n$, the fractional ice concentration in thickness category $n$ (there is also an open water fraction $a_0$ ). The five thickness categories $h_n$ we use have lower bounds (in metres) of 0, 0.6, 1.4, 2.4 and 3.6. For this study we have thickness distribution observations derived from the individual observations of thickness from CS2. These individual measurements are binned according to the thickness distribution used in our CICE model (Schröder et al., 2019), with measurements over one month used to find monthly mean values. We then have observations of ten different variables; $a_n^*$, the area of ice (as a proportion of the total ice) in categories 1-5, where the open water fraction $a_0$ in that grid cell is unknown, and $h_n$, the mean thickness of ice in categories 1-5. For $a_n^*$ this means

$$\sum_{n=1}^{5} a_n^* = 1.$$

These observed variables are related to the state variables $a_n$ and $v_n$ by

$$a_n = a_n^* a$$
$$h_n = \frac{v_n}{a_n},$$

where $a$ is the total fraction of sea ice in a grid cell, $a_n$ is the fraction of sea ice in thickness category n in a grid cell and $v_n$ is the volume of ice per unit grid cell area in category $n$. Precise error statistics on these measurements are difficult to derive due to the nature of the sea ice cover and the multiple sources of contributing errors. To find an error approximately consistent to the error used in the CS2 mean thickness we do some error analysis shown in Appendix A. It is difficult to find errors for ice





concentration and ice thickness in each category that are consistent with all estimated values of mean ice thickness (which used relative errors). However we find that using a total error of 0.3 for ice concentration in each category and 0.8 m for ice thickness in each category leads at least to errors that are close to (or slightly worse than) the mean ice thickness error equivalent.

In this paper we assimilate categories 1-4 in the thickness distribution alongside the mean thickness. Since the sum of the
elements of the thickness distribution is constrained, errors must be correlated between elements. Although we have observations in all ice thickness categories, instead of assimilating observations of $a_n^*$ and $h_n$ for $1 \leq n \leq 5$, with the approximation of uncorrelated errors, we assimilate these observations only for $1 \leq n \leq 4$, plus the mean thickness. Assuming that errors in the latter data are uncorrelated is presumably a less damaging approximation than the first. In this paper, reference to the assimilation of observations of the thickness distribution is shorthand for the latter collection of data.

## 4 Results

### 4.1 Experiment Setup

The ensemble members are generated from a single 1979-2011 stand-alone CICE run of the same configuration as the CICE-PDAF version. From this run we use the model state on January 1st 2012 to initialise all 100 ensemble members, and CICE-PDAF then runs in ensemble mode for a year with no assimilation to generate some initial ensemble spread using the perturbed
atmospheric forcing. Assimilation starts at the beginning of 2012 for either one or four years depending on the experiment. We performed a number of different sea ice DA experiments in order to optimise and fine-tune some of the EnKF settings and to assess the performance of the assimilation. In Table 2 we describe the configurations of these runs. In all experiments, sea ice concentration is assimilated daily, and CS2 thickness observations are assimilated monthly, using monthly means and assimilated in the middle of the month, with the model equivalent of the observation constructed using the daily mean on the
day of assimilation. We chose to assimilate these data on a monthly basis because the errors for a daily product will be even more difficult to ascertain and will essentially involve a 'smearing' of the monthly data. Assimilating CS2 data as monthly means does mean that some of the changes in the sea ice model can be significant, with large increments in grid cells where the model has a high degree of ensemble spread and its ensemble mean differs significantly from the observations. We found that assimilating only once a month outside of the melt season (October-April) still enabled the model to retain significant
information from the assimilation. We also found that assimilating at the end of the month had a negative effect on the system's estimates of sea ice in the next month. Although total sea ice concentration and mean sea ice thickness are not in the CICE state vector, their assimilation has a large affect on the CICE state vector through the correlations between each variable. For example, when assimilating sea ice concentration, the ice concentrations in individual categories will be affected by their correlation with the total sea ice concentration calculated by the observation operator. This means that even when not assimilating
sea ice thickness, the assimilation can still have an effect on the model estimates of sea ice thickness (and potentially all other variables in the CICE state vector).





**Table 2** Configurations of different CICE-PDAF experiment runs. SIC indicates assimilation of daily sea ice concentration (NASA Bootstrap), SIT indicates assimilation of monthly mean sea ice thickness from CS2, and SID indicates assimilation
of monthly mean sea ice thickness distribution from CS2. $n_e$ indicates the number of ensemble members in the run, ff is the forgetting factor $\rho$, $r_l$ is the localisation radius, $\alpha$ is an amplificator factor for the perturbation of the atmospheric forcing fields and $t$ indicates the length of the assimilation run in years.

| run name | SIC | SIT | SID | $n_e$ | ff ($\rho$) | $r_l$ | $\alpha$ | $t$ |
|---|---|---|---|---|---|---|---|---|
| control | N | N | N | 100 | N/A | N/A | 1 | 4 |
| assim_conc | Y | N | N | 100 | 0.995 | 100 km | 1.5 | 4 |
| assim_conc_hi | Y | Y | N | 100 | 0.995 | 100 km | 1.5 | 4 |
| assim_conc_hi_f100 | Y | Y | N | 100 | 1.00 | 100 km | 1.5 | 1 |
| assim_conc_hi_f99 | Y | Y | N | 100 | 0.99 | 100 km | 1.5 | 1 |
| assim_conc_hi_f98 | Y | Y | N | 100 | 0.98 | 100 km | 1.5 | 1 |
| assim_conc_hi_loc200 | Y | Y | N | 100 | 0.995 | 200 km | 1.5 | 1 |
| assim_conc_hi_loc400 | Y | Y | N | 100 | 0.995 | 200 km | 1.5 | 1 |
| assim_conc_hi_loc50 | Y | Y | N | 100 | 0.995 | 50 km | 1.5 | 1 |
| assim_conc_hi_amp1 | Y | Y | N | 100 | 0.995 | 100 km | 1 | 1 |
| assim_conc_hi_amp2 | Y | Y | N | 100 | 0.995 | 100 km | 2 | 1 |
| assim_conc_hi_4hd | Y | N | Y | 100 | 0.995 | 100 km | 1.5 | 4 |


### 4.2 Tuning the assimilation settings

PDAF provides some assimilation settings that can be tuned for the EnKF to function as the user wishes. In the LETKF, there are three important factors to consider. These are the ensemble size, forgetting factor and localisation radius. For the ensemble
size, we want to use as large an ensemble as possible to reduce the need for localisation and inflation. Therefore we use an ensemble size of 100. Figure 1 shows the effect of the forgetting factor on the ensemble spread of sea ice volume. The forgetting factor, like ensemble size, will work alongside the atmospheric forcing perturbations to increase ensemble spread and reduce the likelihood of significant undersampling and ensemble collapse. We have generally found that the system is very sensitive to changes in the forgetting factor, particularly when we assimilate CS2 products. In Fig. 1 we show some one-year assimilations
with different forgetting factors (0.98, 0.99, 0.995 and 1.00). As we expect, decreasing the forgetting factor will increase the ensemble spread of the experiment. We found that using these low forgetting factors increases the chances of the sea ice model crashing. These crashes are related to unphysical states being formed such as 'ice spikes', where ice in grid cells (usually close to the ice edge) can have very small concentrations but thicknesses of well over 10 m. This problem often causes the model to crash, and although fixable in post-processing becomes worse as the forgetting factor is lowered. Using 0.995 does not lead to



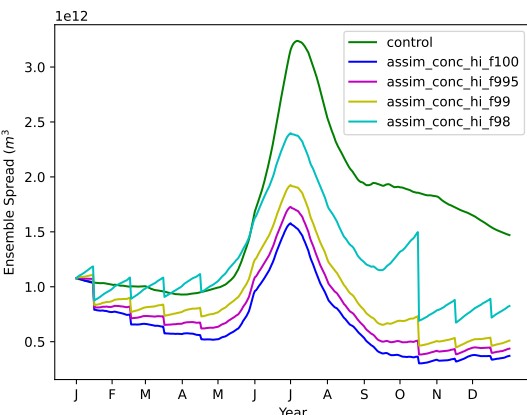

**Figure 1.** Sea ice volume ensemble spread (one standard deviation) for 2012 in a control run and four CICE-PDAF runs using forgetting factors of 1.00, 0.995, 0.99 and 0.98.

any model crashing, and this is the value used in all future experiments.

We also perform experiments with localisation radii between 50 and 400 km. Our choice of model grid means that without varying the localisation too much or choosing unreasonable localisation parameters an observation will only maximally affect

grid cells which are 8-10 grid cells away. Grid cells located at the sea ice edge are most likely to be affected by these changes in localisation. In Fig. 2 we show Pan-Arctic maps of sea ice thickness in March and October for these runs with different localisation parameters. The changes in the results when using differing localisations only show very slight differences with one another. We settle for using a 100 km localisation radii as a compromise between smaller radii close to coastlines and larger radii in the middle of the ice pack.


Finally we look at the potential in amplifying the perturbations to the atmospheric forcing fields, used to generate ensemble spread as mentioned in Sect. 2.3. In Fig. 3 we show the spread in the sea ice volume for the control and three assimilation runs, with the only difference in the assimilation runs being a change in the amplification of the perturbations ($\alpha = 1.0, 1.5, 2.0$). As

expected, amplifying the perturbations increases the ensemble spread in the sea ice, particularly in summer. In some cases with $a = 2$ the assimilation runs have a higher spread than the control. However we have found that amplifying the perturbations this much can cause the model to become unstable, causing crashes in some of the ensemble members from a wide variety of issues. Due to this we only use $a = 1.5$, which does not cause these issues but still increases ensemble spread significantly. This is a substantial amount of additional ensemble spread, but due to the dissipative nature of the model we want a spread as

large as possible without the model crashing.



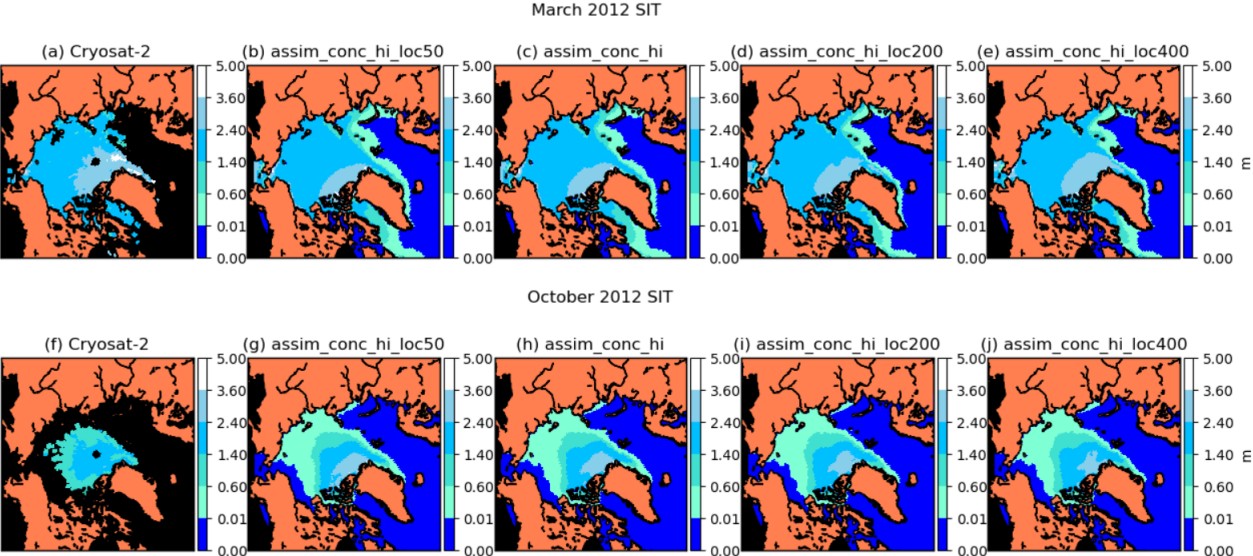

**Figure 2.** Maps of sea ice thickness (m) in March 2012 (top row) and October 2012 (second row). Columns show Cryosat-2 and three CICE-PDAF runs using localisations of 50 km, 100 km, 200 km and 400 km. In this and other maps in this paper, grid cells lacking Cryosat-2 data are shown in black in the Cryosat-2 plots.

### 4.3 Grid Cell Level Analysis

To understand how the assimilation of the thickness distribution will effect changes in the model, we will look at the correlations between some of the model state variables such as sea ice concentration, thickness and volume, in a few grid cells before and after assimilation. We will also look at the evolution of these variables throughout the first year of the assimilation time period. Figure 4 shows the correlations in winter and summer (January 1st and July 1st, 2012) at 2 different grid cells in the Arctic; one in the Fram Strait and one in the Central Arctic region (close to the North Pole). We have chosen these locations

because the concentration, thickness, volume and thickness distribution of the sea ice in each of these will be significantly different because of how the Arctic sea ice cover evolves differently in these regions.

In the Fram Strait the ice cover is seasonal and thin, whereas in the Central Arctic the ice is thicker. In the Fram Strait in winter we see strong negative correlations between thickness and total volume with ice in the first and second thickness cat-

egories. There are positive correlations in the other three categories, with the strength of these correlations increasing with thickness category. This is because this grid cell is mostly covered with ice in the thinnest categories, with small amounts in





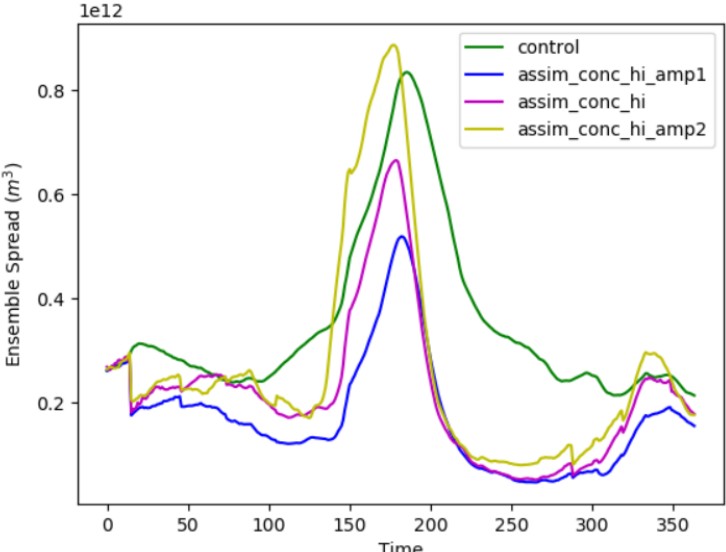

**Figure 3.** Daily sea ice volume ensemble spread (one standard deviation) for 2012 in a control run and three CICE-PDAF runs using atmospheric forcings which have been perturbed with different amplification factors (1, 1.5 and 2).

the higher categories, so relatively small increases in ice in these thicker categories of ice will lead to more volume. Total sea ice concentration is positively correlated with the concentration in the second and third categories at this time and negatively correlated with the others. During summer the Fram Strait correlations look different, because this region is now most likely covered with ice only in the thinnest category, meaning that any small increase of ice in the thicker categories will lead to higher thickness and volume.

In the Central Arctic, where the ice is much thicker (particularly in winter), the correlations between ice volumes in the two thickest categories of ice and the total sea ice thickness and volume are very strong in both winter and summer. There are strong negative correlations with the thinnest two categories. This pattern occurs for both the winter and summer, though in winter there is also a positive correlation between category 3, volume and thickness, which is negative in summer. The correlations between the ice concentration and volume between categories are strong in both seasons, but particularly in summer. As the Central Arctic is covered by 100% sea ice in winter, the correlations between the total sea ice concentration and the ice concentrations of each thickness category are generally smaller than the correlations to the thickness or volume. In summer the correlations are larger, but still not as strong as those in the Fram Strait. Overall this means that if assimilation results in an increase in ice thickness in the Central Arctic grid cell, the sea ice concentration and volume in the thicker categories will be increased, with some decrease in ice in the thinner categories. A common trait of these correlation matrices is neighbouring blocks acting almost as one category, this is because the ice concentration (aice) and volume (vice) in each category have similar autocorrelations and are cross-correlated. When neighbouring categories are negatively correlated there is likely to be



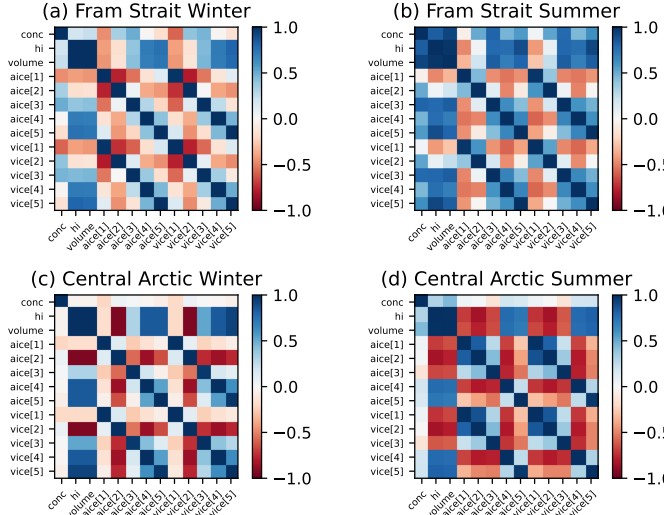

**Figure 4.** Correlation matrices of CICE state variables: fraction of ice in category $n$ (aice[n]), volume of ice per unit grid cell in category n (vice[n]), as well as the total sea ice concentration (conc), grid cell mean ice thickness (hi) and total grid cell sea ice volume (volume) in January 2012 and July 2012 in the Fram Strait and the Central Arctic.

exchange of ice area and volume between them.

In Fig. 5 we show sea ice concentration and thickness in 2012 in the same two grid cells as discussed above. The CICE-PDAF assimilation experiments tend to follow the Bootstrap sea ice concentration observations in a smoothed manner for most of

the year in both of the chosen grid cells compared to the control run. We can see for the assimilation runs that the decrease in concentration in late August in the Fram Strait leads to a remarkable increase in the sea ice thickness at the same time in these runs. The stronger reduction of sea ice concentration in the model vs. Bootstrap causes in artificial increase of ice thickness in the assimilation runs due to the negative correlation between concentration and thickness. The three CICE-PDAF assimilation runs have a much higher mean thickness in this grid cell than the control run, even in the summer months when no CS2 thick-

ness data are assimilated. In the Central Arctic grid cell there is only a decrease in ice concentration in the control run, which is not matched by the observations or any of the assimilation experiments. In assim_conc, sea ice thickness is overestimated in comparison to the CS2 measurements, and is much higher than the other CICE-PDAF runs. The decreases in concentration caused by the assimilation in late summer are not causing a decrease in ice thickness. This could be caused by the covariances between total sea ice concentration and the CICE state variables being very weak during this time of the year. This would mean

that the small analysis updates in the total sea ice concentration might not affect the mean sea ice thickness. In the Central Arctic assim_conc_hi_4hd performs best between October and December but not as well as the other assimilation experiments in the first four months of the year.





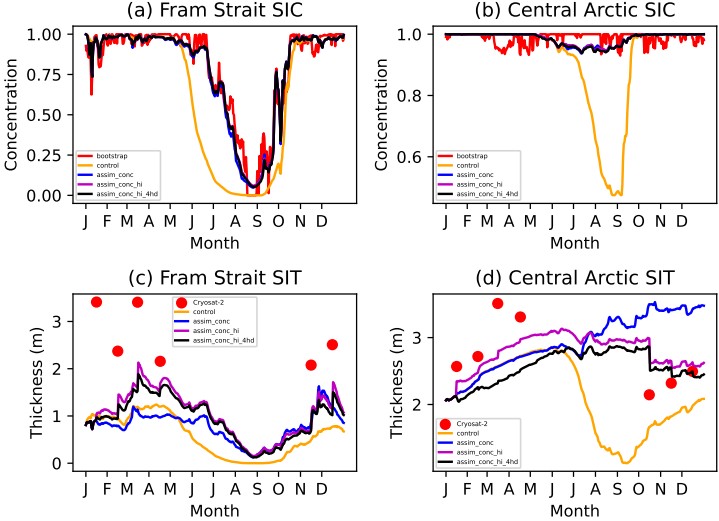

**Figure 5.** Sea ice concentration and mean sea ice thickness in one grid cell in the Fram Strait and the Central Arctic in CICE-PDAF, Bootstrap and Cryosat-2 in 2012.


### 4.4 Evaluation of CICE-PDAF against Bootstrap and Cryosat-2 observations

To assess the effectiveness of the DA we look at the root-mean-square-error (RMSE) of the control runs, assimilation runs and observations assimilated, against the Bootstrap data and the CS2 data randomly selected for validation throughout the four years. In Fig. 6 we show the RMSE in daily sea ice extent for CICE-PDAF against the Bootstrap extent. We see a large reduction in total sea ice extent RMSE when assimilating sea ice concentration (blue, magenta, and gold lines) in comparison to the control run (green line) in all three experiments year-round. Assimilating ice thickness or ice thickness distribution in addition to concentration does not have any visible impact on sea ice concentration.

In Fig. 7 we show mean thickness in CICE-PDAF against CS2 for our four experiments. Assimilating CS2 mean sea ice thickness significantly reduced the RMSE in ice thickness compared to the evaluation data. We can see that the control has a significant number of data points which lie outside the dense central region. For the assim_conc experiment, the ice appears to be much thicker than in Cryosat-2 . With the assimilation of mean thickness included in assim_conc_hi the central region following the linear trend is now more visible. This is also reproduced in assim_conc_hi_4hd, though there is a bulge





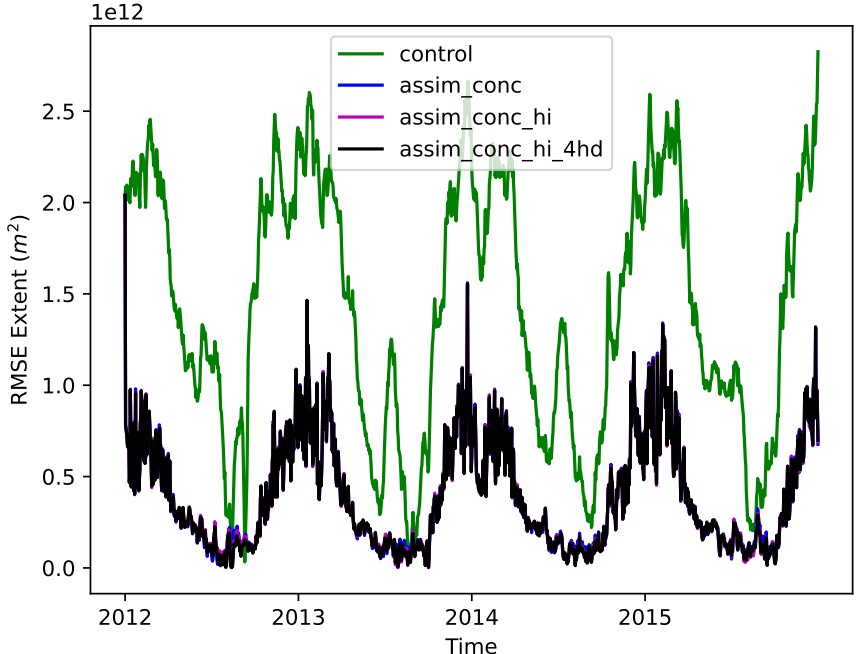

**Figure 6.** RMSE of daily sea ice extent for the control and three assimilation runs of CICE-PDAF in comparison to the Bootstrap extent from 2012 to 2015.

of slightly thicker ice in CICE-PDAF between 1.5 and 2.5 m. In general it appears that the model is weighted towards slightly thinner ice than the equivalent from Cryosat-2 observations. It seems that most of the improvement in thickness estimates when assimilating Cryosat-2 data comes from removing a substantial quantity of the thickness estimates in the control that are too high. Although the high density of low thickness estimates between 1 and 2 m that appear in the 2d histogram plot for control compared to that in assim_conc_hi and assim_conc_hi_4hd is smaller the assimilation of the products is not able to completely remove this bias.

In Table 3 we show the RMSE for mean thickness and category 5 thickness compared to CS2. The RMSE in mean thickness was more than halved when assimilating CS2 sea ice thickness in comparison to the control run from 0.63 m to 0.27 m. On the other hand assimilating Bootstrap concentration alone appears to be detrimental to the model estimates of monthly mean thickness, as shown by the assim_conc experiment, which had a higher RMSE (0.88 m) than the control. As shown by the 2d histogram plot in Fig. 7 (top right) this seems to be caused by the assimilation of concentration as the model estimates tend to be much thicker than in CS2. In terms of the mean thickness, assimilating the ice thickness and concentration in the four thinner categories of ice did not improve sea ice thickness estimates. However assim_conc_hi_4hd did lead to significantly




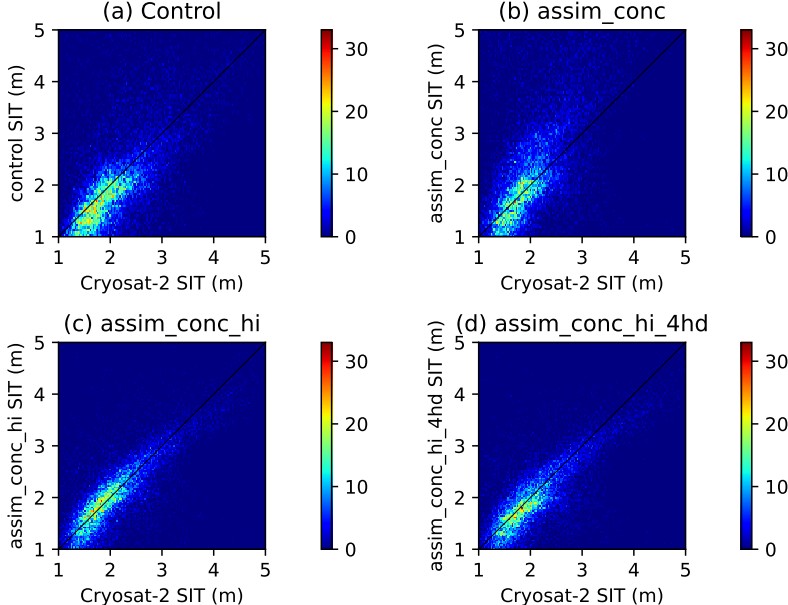

**Figure 7.** 2d histogram plots of sea ice thickness estimates in CICE-PDAF against the Cryosat-2 evaluation data for four CICE-PDAF experiments.

improved estimates of the thickness in category 5 (RMSE of 1.7 m vs. 3.2 m), even though information about category 5 was not directly assimilated (see Sect. 3.3). This improvement is not seen in any of the other CICE-PDAF runs. There were no changes to the RMSE in the thickness or concentration in the other four categories compared to the CS2 data. This change in thickness in category 5 also has a generally positive effect on the model estimates of volume in this category.

**Table 3** RMSE of the domain-averaged monthly mean ice thickness (hi) and ice thickness in category 5 (hice5) to Cryosat-2 evaluation data

| RMSE | hi | hice5 |
|---|---|---|
| **control** | 0.62 | 3.2 |
| **assim_conc** | 0.88 | 3.3 |
| **assim_conc_hi** | 0.27 | 3.3 |
| **assim_conc_4hd** | 0.27 | 1.7 |

In Fig. 8 we show monthly mean RMSE of ice thickness in category 5 for each experiment, compared to CS2, and snapshots of how this difference in thickness for one month (December 2013) leads to an improvement in the volume in this category (using Bootstrap data to convert CS2 thickness estimates into volume). We found that additionally assimilating thick-





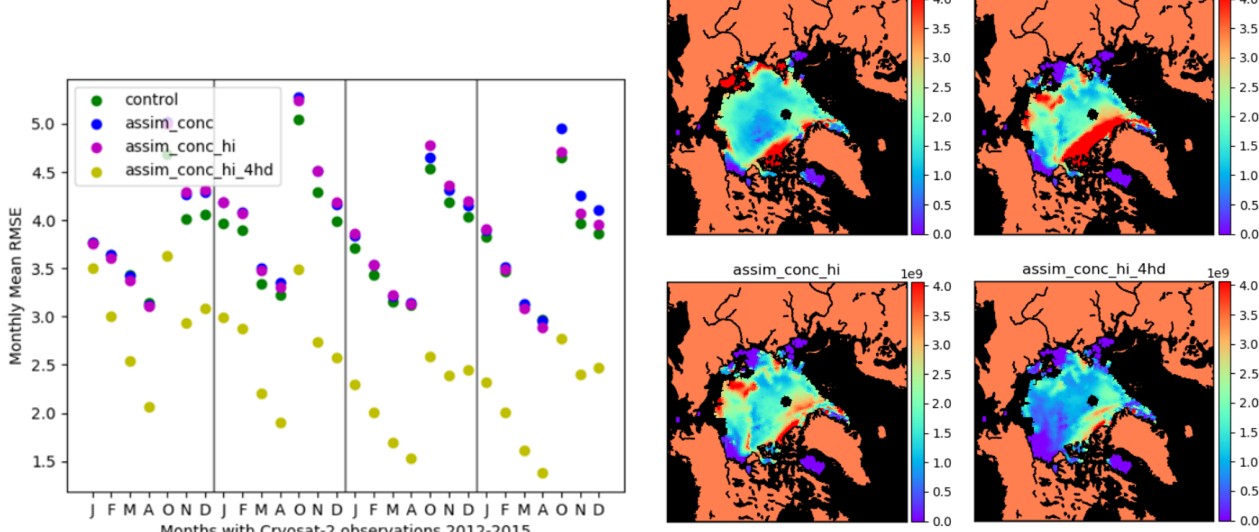

**Figure 8.** LHS: RMSE of Category 5 sea ice thickness between 2012 and 2015 for January, February, March, April, October, November and December for the control and CICE-PDAF assimilation runs, against Cryosat-2 sea ice thickness evaluation data. RHS: Maps of category 5 sea ice volume RMSE in December 2013 for four CICE-PDAF runs against Cryosat-2 evaluation data. Clockwise from the top left map: control run, assim_conc, assim_conc_hi_4hd and assim_conc_hi.

ness distribution leads to a significant decrease in the thickness and volume RMSE of category 5 in the model, because CS2 estimates of thickness in this category are generally much lower. Assimilation of concentration and mean thickness seemed to

have a small negative effect on the estimates of category 5 thickness and volume for most of the experiment in comparison to the control. The improvement in category 5 RMSE occurs because the category 5 ice thickness in the control model is 6 to 7 metres in most places year round, whereas CS2 has much thinner estimates around 3.5 to 4 metres. In many areas the thickness and volume of ice of the thickest category is much higher than in CS2, and only assimilating part of the thickness distribution seems to be able to resolve this difference. The assimilation of the concentration and thickness in the lowest four categories

then has an important effect on achieving better estimates of the thickness, volume and mass budget distribution in comparison to observations on a pan-Arctic scale.

**4.5  CICE-PDAF Assimilation Results**

Looking at time series of daily pan-Arctic sea ice area, Fig. 9, we see that in all four years, the sea ice area in the freeze-up season is smaller, and in the melting season is larger than the control, leading to a narrower seasonal variation. On a pan-Arctic





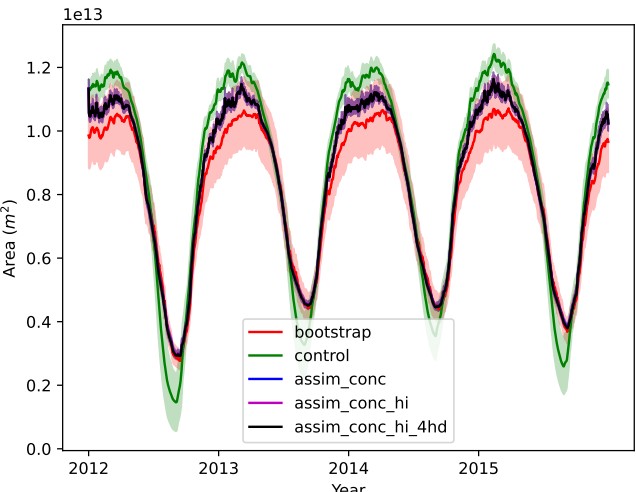

**Figure 9.** Pan-Arctic sea ice extent from 2012-2015 from Bootstrap observations and four different CICE-PDAF runs. Solid lines show observations or ensemble mean and the shading shows observation error or ensemble spread (one standard deviation)

scale we see again that the assimilation of the concentration is the key factor for this area diagnostic, with the assimilation of CS2 products providing no additional benefits, as expected.


We show pan-Arctic sea ice volume in Fig. 10. We see that the assimilation of only concentration (blue) increases sea ice volume over the experimental time frame in comparison to the control run (green). The assimilation of monthly mean thickness in addition to this (magenta and gold) appears to mitigate this effect, where assim_conc_hi (magenta) has sea ice volumes in

winter comparable to the control run, but on the other hand it has greater sea ice volumes in summer. The assimilation of the sub-grid scale thickness distribution (gold) appears to cause a rebalancing of the ice thicknesses and fractional areas in each category, leading to large swathes of ice being moved into lower thickness categories, which results in lower sea ice volumes. A further analysis (see Fig. 12) shows that this is caused by large differences between the CS2 ice thickness in category 5 and the model ice thickness in category 5. In the first four months of the experiment the increments in sea ice volume caused by

the thickness distribution assimilation are significant because of these differences.

In Fig. 11 we show maps of sea ice thickness in October 2012-2015 from CS2, the control run, three CICE-PDAF experiments and PIOMAS. We can see that CS2 has thinner sea ice than all the CICE-PDAF experiments and PIOMAS during this

time period. In particular the control and assim_conc experiments show a tendency to pile up thicker and thicker ice against the Canadian Archipelago. If we compare the ice thickness in assim_conc to that in assim_conc_hi we can see that the assimilation





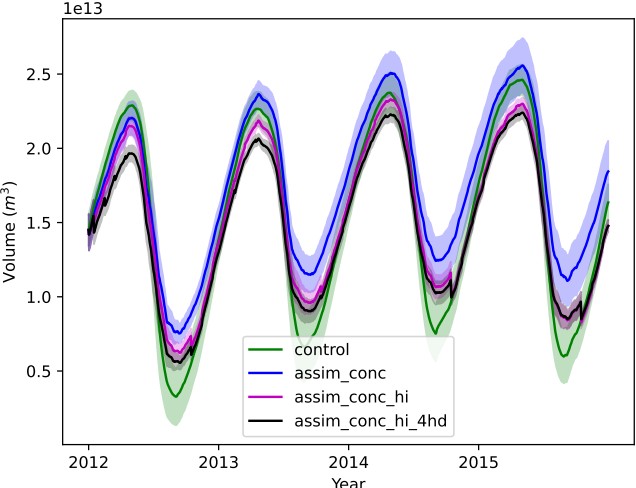

**Figure 10.** Pan-Arctic sea ice volume from 2012-2015 in four different CICE-PDAF runs. Solid lines show observations or ensemble mean and the shading shows observation error or ensemble spread (one standard deviation).

of the mean thickness alongside the ice concentration significantly reduces the gradient of sea ice thickness from the sea ice edge towards the Canadian Archipelago. The assim_conc_hi experiment has much thinner ice in these regions, similar to CS2. In assim_conc_hi there is instead a much more homogeneous (in terms of mean thickness) ice cover across the Arctic, with a

majority of the ice cover having a mean ice thickness in October between 1.4 and 2.4 metres. The assimilation of the CS2 mean ice thickness appears to be very effective in CICE-PDAF, with the ice thickness of assim_conc_hi looking quite similar to that of the CS2 product, and shown by the reduction in RMSE in Table 3. The assimilation of the concentration does however have the effect of significantly reducing the amount of thin ice at the edges of the sea ice pack, with much less ice between 0 and 0.6 metres in the Arctic in October in the CICE-PDAF experiments. Additionally assimilating the four thinnest categories of

sea ice in the sub-grid scale thickness distribution generally had the effect of slightly increasing the thickness of the thickest ice, however this still has very good agreement with CS2. The assimilation of the sub-grid scale thickness distribution is also most similar to PIOMAS at this time. The assimilation of the CS2 products from October through April causes a thicker sea ice cover to persist throughout the summer period than we see in the control model, with higher sea ice extents and volumes in summer (not shown).


Alongside ice thickness, we also look at sea ice concentration in September 2012-2015 in Fig. 12. Here all runs with assimilation of sea ice concentration showed very similar results, and similar to the Bootstrap sea ice concentration. This shows what the RMSE was telling us from the top panels in Fig. 6 – that the assimilation of sea ice concentration works well, and

moves the concentration estimates in the assimilation experiments close to the observations. A significantly smaller marginal



**Figure 11.** Monthly mean sea ice thickness (in metres) in October 2012, 2013, 2014 and 2015 in Cryosat-2 and our CICE-PDAF experiments (ensemble mean) and PIOMAS.





ice zone (MIZ, the area of ice containing between 0.15 and 0.8 sea ice concentration), which is present in the Bootstrap sea ice concentration, is also seen in the assimilation runs. The Bootstrap concentration tends to feature ice concentrations close to 1 inside the ice pack, which could also lead to thicker ice within the ice pack because higher ice concentrations are expected to correlate with higher ice thicknesses – in Fig. 4 strong correlations are shown between concentration and thickness except for

the Fram Strait in Summer. These wide areas of high concentrations in the ice pack are not present in other sea ice concentration observation products such as the NASA team product. For this reason it would be interesting to compare the assimilation of different sea ice concentration products.

In Fig. 13 we show sea ice volume in each thickness category and the total Arctic sea ice volume for 2012-2015. Large differences occur in the three largest categories, which we expect because smaller changes in the amount of ice in these categories would lead to bigger differences in their volume. In the thinnest category (ice 0.6 m and thinner), there are small decreases in the amount of ice in this category when compared to the control, but the three assimilation runs are generally similar to one another. The largest differences seem to occur at the end of each melting season, with the control run having significantly more

ice in the thinner categories than the assimilation experiments. In category 2 (0.6-1.4 m) the same pattern occurs, but shifted slightly (with the maximum ice volume in this category occurring around late December and early January), and again with significantly more ice at this time in the control run.

In category 3 (ice thickness between 1.4-2.4 m) we first see larger differences between the assimilation runs. Most notably

the assimilation of the thickness distribution (assim_conc_hi_4hd) has caused a significant decrease in ice in this category compared to assim_conc_hi at the end of the freezing-up season (March-April) in 2013. This decrease persists throughout the rest of the run. The ice in this category in the control run tends to experience the greatest variation between the minimum and the maximum, generally having the lowest amount of ice in this category at the end of summer but the most at the end of winter. In assim_conc the reverse is true. The assimilation of thickness alongside concentration has a substantial effect on the

volume of ice in this category, particularly during the minimum, with assim_conc_hi having much larger ice volume estimates in this category than assim_conc. Additionally assimilating sub-grid scale thickness distribution seemed to remove this category 3 ice created by the mean thickness assimilation, with assim_conc_hi_4hd having similar category 3 volume estimates to assim_conc. In category 4 (2.4-3.6 m), assimilation leads to more ice in this category, particularly when assimilating the sub-grid scale thickness distribution, where the analysis increments are quite large. This increase of ice volume in category 4

seems to counteract the decrease in categories 3 and 5 in this run.

Looking at the thickest ice (3.6 m and thicker), the assimilation of concentration alone causes a large increase in category 5 volume. The assimilation of mean thickness tends to decrease the volume of the thickest ice compared to the control. The addition of sub-grid scale thickness distribution assimilation does not seem to significantly affect the total volume of ice in this

category, however regional differences could be significant. It seems that the primary reason for the decrease in sea ice volume







**Figure 12.** Monthly mean sea ice concentration in September 2012, 2013, 2014 and 2015 in the Bootstrap observations, the control and three CICE-PDAF experiments (ensemble mean).





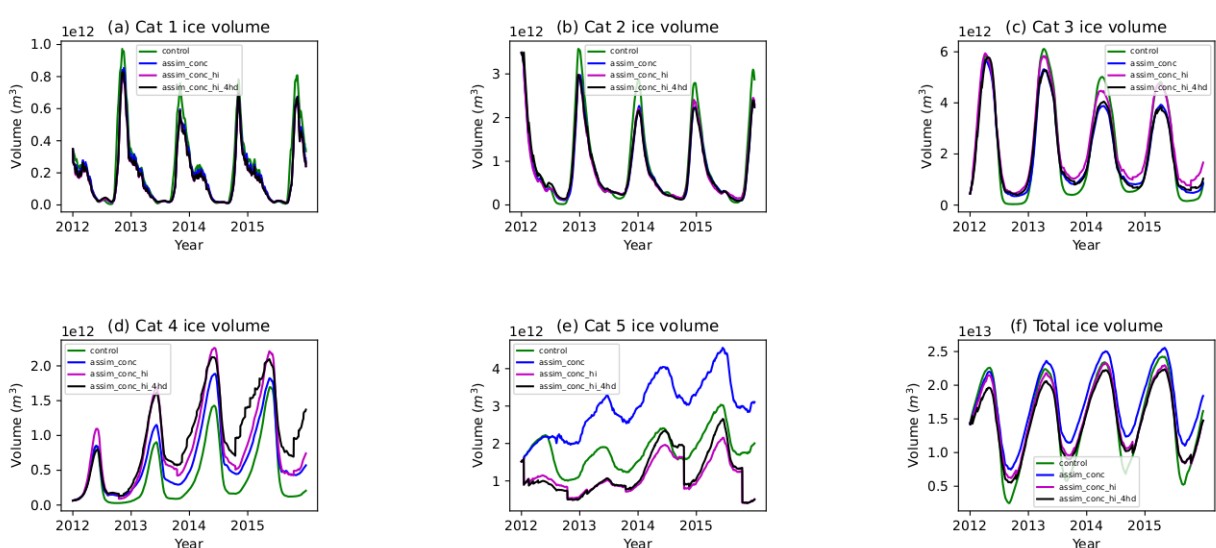

**Figure 13.** Volume of ice in each thickness category and the total volume for each of the CICE-PDAF experiments (ensemble mean).

when assimilating thickness distribution is that a decrease in ice in categories 3 and 5 is partially (but not fully) counteracted by an increase in category 4 ice volume. Overall we see that assimilation of the thickness distribution product can have wide-reaching impacts on the distribution of the sea ice in the Arctic.


## 5 Discussion

Overall, the assimilation of sea ice observations decreased the seasonal variation of both sea ice extent and sea ice volume in the freeze-up and melting periods in comparison to the control run. The assimilation of the CS2 observations resulted in an
increase in the area of thicker ice extending outwards to the North pole from the Canadian Archipelago, but was compensated by a reduction in the thickness in the regions of the thickest ice. The sea ice in the control experiment was generally too thin throughout most of the year, except in the Canadian Archipelago region. As a lot of thin first year ice is being formed in the control model this will make it more susceptible to subsequent melting during the melting season, and hence increase the seasonal variation. The model generally appears to overestimate both summer ice melt and the winter freeze up, except in
the Canadian Archipelago where the ice gets much thicker than observations from CS2. This may be caused by too much ice advecting into this region in the model or the ridging formulation in the model favouring the formation of too much thick ice. This problem could be being exacerbated by possible issues with the climatology we use to simulate the oceans, which may



cause too much congelation growth in this region.

Apart from in assim_conc_hi_4hd, the estimates of ice in the thickest category are poor (see Fig. 8), so the model appears to have significant difficulties in simulating the thickest sea ice. In assim_conc the ice thickness and volumes are substantially larger than in the control, particularly in the regions of the thickest ice (ice thickness > 3.6 m). This could indicate issues with the open water fraction in the CICE model as CICE generally has lower sea ice concentrations within the ice pack than Bootstrap (which has concentrations generally close to 1 away from the ice edge) and the correlations between concentration

and thickness within the model could cause new ice created to be thicker than it should be. The marginal ice zone (MIZ) which is defined as the area of ice between 15% and 80% concentration, forms a boundary region of small sized ice floes between the open ocean and the central ice pack which are strongly affected by ocean waves (Sundfjord et al., 2007). The interactions between the atmosphere, ocean and sea ice are particularly strong in the MIZ, and a reduction in MIZ could also have strong impacts on Arctic sea ice ecosystems (Barber et al., 2015). The assimilation of Bootstrap sea ice concentration also appears

to decrease the marginal ice zone (MIZ) area, which is again an artefact of the characteristics of the Bootstrap sea ice concentration. A wholesale change in the mean thickness across the Arctic will change how a lot of this ice behaves because the dominant physical processes on the sea ice change depending on its thickness. As the ice is much thicker the ocean waves have a much smaller effect on it. Thicker ice created by the assimilation can also advect into the Beaufort and Lincoln seas causing higher than expected ice thickness and volume in these regions.


The areas of thickest ice (> 3.6 m) and thinnest ice (< 0.6 m) narrowed rather than broadened when ice thickness products from CS2 were additionally assimilated. This resulted in a significant increase in the area of ice between 0.6 and 3.6 m thick. In addition to the effects of the sea ice concentration assimilation, this could partially explain why the sea ice extent minima in September were larger in the assimilation runs than the control model every year, because the thicker ice would be more

resilient to being fully melted during the melt season than the thinner ice. This study highlights significant benefits of observations of sea ice thickness and sub-grid scale thickness distribution for estimation of sea ice thickness and volume using data assimilation. It shows that there should be further emphasis put on making future observations of sea ice thickness in order to establish a more accurate long term record of ice thickness and volume in a reanalysis, and also in making use of different types of observations in sea ice data assimilation studies to ascertain a clearer picture of the Arctic sea ice.


One shortcoming of this study is that the observation errors of ice thickness, and the sub-grid scale thickness distribution (ice concentration and ice thickness in each category) are highly correlated, but for simplicity we do not account for observation error correlations. The observation error statistics on the sub-grid scale thickness distribution and the mean thickness are both highly uncertain and are assimilated only once a month and only outside of the Arctic melting season. Assimilating only

once a month can cause large increments in some of the CICE state variables especially in October, after the 5 month Summer period without assimilation. The assim_conc experiment differs significantly from assim_conc_hi and assim_conc_hi_4hd so the thickness assimilation is not only highly effective in reducing the RMSE in mean ice thickness but has a significant effect in



September, 5 months after any thickness or thickness distribution assimilation has taken place. A more accurate assessment of the uncertainties in the Cryosat-2 products assimilated in this study would likely lead to significant improvement in the system

and estimated sea ice state.

Another important factor to consider is that many of the variables in the CICE state vector are assumed to be Gaussian are bounded, for example sea ice concentration, as well as many other state variables which have upper or lower bounds. This means that we need to account for the correlations in the LETKF leading to unphysical sea ice states, so some variables need to

be altered after the assimilation to avoid model crashes. Of the observations we assimilate, sea ice concentration and sub-grid scale thickness distribution have upper and lower bounds, and sea ice thickness has only a lower bound, therefore the observation error covariances for our observations should approach zero as the bounds are reached (Bishop, 2019). For simplicity we have not applied this in our study. The observation error variances we have used for sea ice concentration and sea ice thickness do depend on the measured state, and in the case of sea ice concentration do get smaller as some bounds are approached, but

this is done to account for uncertainties in the observational method. To evaluate the ice thickness from the studies in this chapter, we used randomly selected Cryosat-2 data held back for evaluation. When evaluating the studies with the Cryosat-2 we have to recognise that although the data held back for evaluation was randomly selected, it is not completely independent and so the evaluation data will be correlated in some way with the observations chosen for assimilation.

Assimilating Bootstrap sea ice concentration alone resulted in an increase in sea ice thickness and volume in comparison to the control experiment, and it performed worse against our evaluation CS2 dataset. This effect has not been seen seen in other studies assimilating only ice concentration and evaluating modelled ice thickness. In Fritzner et al. (2019) when only OSISAF sea ice concentration was assimilated estimates of thickness generally improved in comparison to independent observations, only performing worse in May, and unlike in our experiment the ice thickness and volume were generally reduced. It

appears to be caused by the assimilation of Bootstrap sea ice concentrations in the summer months. There are a few months in summer when Bootstrap sea ice concentrations are higher than the control model, which would lead to positive increments in sea ice concentrations. This may be causing unrealistic thickening of the sea ice through positive correlations between sea ice concentration and sea ice thickness within the data assimilation. It seems that further care is required when assimilating sea ice concentration alone when using assimilation schemes like we have used in this study. This is not an issue for other reanalyses

like PIOMAS, which uses a highly tuned optimal interpolation technique which only affects ice concentration close to the ice edge, and assimilates a different dataset of sea ice concentration observations (NASA Team). It is likely a result of some unique characteristic of the Bootstrap sea ice concentration in combination with the physics of the CPOM-CICE model and the data assimilation scheme we use. It does show that care should be taken in the interpretation of sea ice volume estimates when assimilating sea ice concentration alone in any reanalysis study.


In this study we attempted to tune a number of important assimilation parameters - forgetting factor, localisation radius and an amplification factor. We attempted to do this using a short 1-year assimilation study, which we assumed is a reasonable length





with which to determine their long term effects. However we showed that changing any of these parameters could in some cases result in a considerably changed estimate of the sea ice state, without altering any part of the model or observations assimilated. There is a significant amount of room for fine-tuning these parameters which could result in changes in the estimates of the sea ice state. The localisation radius study did show that larger radii may slightly improve the model estimates, but these larger radii are probably physically unrealistic for the range of the effects of the CPOM-CICE model sea ice dynamics. In the future a location-dependent localisation radius would be beneficial, especially near the sea ice edge. However they would not likely change any of the outcomes of the study with regards to the intercomparison between each assimilation experiment. With regards to ensemble size, more is generally better for the health of a data assimilation system, though 100 is on par with, for example, TOPAZ4, and much larger than in Fritzner et al. (2019).

In this study we have assimilated a sub-grid scale thickness distribution for the first time. When this is additionally assimilated alongside the mean thickness there are benefits to the estimates of ice in category 5. The model appears to overestimate the thickness of the thickest ice substantially. However it could be argued that this region where category 5 sea ice is present in large quantities is the least important because the ice cover has not changed as drastically here as it has in any other region of the Arctic. The ice is so thick in this region that even under increased Arctic warming the ice concentration here is close to or is 1, and has been throughout the satellite era. However a more accurate sub-grid scale thickness distribution could lead to medium and longer term benefits to sea ice estimates in the rest of the Arctic.

## 6 Conclusions

We have produced a new sea ice data assimilation system CICE-PDAF, using the method of LETKF assimilation with a localisation radius of 100 km, a forgetting factor of 0.995 and an ensemble size of 100. Ensemble spread was further generated in the system by perturbing NCEP-2 atmospheric forcing fields using an EOF method, and amplifying these perturbations to further increase it. Using this DA system we conducted experiments assimilating NASA Bootstrap daily sea ice concentration alongside CS2 products of monthly mean sea ice thickness and monthly mean sub-grid scale sea ice thickness distribution. This is the first time that a sub-grid scale sea ice thickness distribution product has been assimilated.

We have performed experiments comparing a control run (no assimilation) of the CICE-PDAF model alongside assimilation runs and found that the assimilation performed well for sea ice concentration, sea ice thickness and sub-grid scale sea ice thickness distribution. The best performing experiment in comparison to independent observations was assim_conc_hi_4hd, which assimilated sea ice concentration, mean sea ice thickness and the sea ice thickness distribution in the thinnest four categories of sea ice. For the first time we have assimilated a sub-grid scale sea ice thickness distribution, which caused significant changes in the ice thickness distribution across the ice cover. The primary benefit from this was the significantly improved estimates of the ice thickness in the thickest category (a reduction in RMSE of 1.5 m), which persisted throughout the time period. By assimilating thickness distribution alongside mean thickness, there were important benefits to the thickness distribution and





sea ice mass budget estimates in our sea ice model, primarily in the thickest category of sea ice.

Overall we have shown that the assimilation of the CS2 products were highly beneficial when compared to the assimilation of concentration alone against randomly chosen CS2 evaluation data that were not assimilated. The enormous differences in ice

thickness between assimilating only ice concentration and assimilating both ice concentration and ice thickness are striking and suggest that information about sea ice thickness is essential to produce a realistic sea ice reanalysis data set. The assimilation of the Bootstrap concentration in fact appeared to negatively effect CICE-PDAF estimates of sea ice thickness compared to these evaluation data.

*Code and data availability.* CICE v5.1.2 is available here: https://github.com/CICE-Consortium/CICE-svn-trunk/tree/cice-5.1.2. PDAF v2.0

is available for download here: https://pdaf.awi.de/trac/wiki.

## Appendix A: Appendix A

The mean ice thickness of a grid cell is

$$h = \sum_{n=0}^{5} a_n h_n,$$

where $a_n$ is ice concentration and $h_n$ is ice thickness in thickness categories 1-5 (with category 0 as open water). Allowing for errors in all the quantities such that $h = h^{\text{true}} + \epsilon_h$, $a_n = a_n^{\text{true}} + \epsilon_{an}$ and $h_n = h_n^{\text{true}} + \epsilon_{hn}$, where "true" indicates the true values and $\epsilon$ indicates the errors, we find


$$h^{\text{true}} + \epsilon_h = \sum_{n=0}^{5} (a_n^{\text{true}} + \epsilon_{an})(h_n^{\text{true}} + \epsilon_{hn})$$

$$= \sum_{n=0}^{5} \left( a_n^{\text{true}} v_n^{\text{true}} + a_n^{\text{true}} \epsilon_{hn} + h_n^{\text{true}} \epsilon_{an} + \epsilon_{an} \epsilon_{hn} \right).$$


Since $h^{\text{true}} = \sum a_n^{\text{true}} v_n^{\text{true}}$, this reduces to

$$\epsilon_h = \sum_{n=0}^{5} \left( a_n^{\text{true}} \epsilon_{hn} + h_n^{\text{true}} \epsilon_{an} + \epsilon_{an} \epsilon_{hn} \right).$$






We do not know these errors, but we do assume we know their root mean square expected values $\overline{\epsilon_x^2} = \sigma_x^2$, where the overline indicates average over realisations. Neglecting the triple and quadruple products leaves us with

$$\overline{\epsilon_h^2} = \sum_{n=0}^{5} \sum_{n'=0}^{5} \left( a_n^{\text{true}} a_{n'}^{\text{true}} \overline{\epsilon_{hn}\epsilon_{hn'}} + a_n^{\text{true}} h_{n'}^{\text{true}} \overline{\epsilon_{hn}\epsilon_{an'}} + h_n^{\text{true}} a_{n'}^{\text{true}} \overline{\epsilon_{an}\epsilon_{hn'}} + h_n^{\text{true}} h_{n'}^{\text{true}} \overline{\epsilon_{an}\epsilon_{an'}} \right).$$

To simplify this and be able to use it to make an estimation of the expected error statistics we make two assumptions, firstly that the errors in area and height are uncorrelated, and secondly that the errors of ice concentration and thickness are uncorrelated between categories. This is likely to be untrue, especially considering that the sum of the ice concentrations in each
category are bounded, but without these assumptions it would be difficult to approximate the error statistics in the assimilated ice concentration and thickness in each category. With these assumptions we find

$$\sigma_h^2 = \sum_{n=0}^{5} \left( a_n^{\text{true}^2} \sigma_{hn}^2 + h_n^{\text{true}^2} \sigma_{an}^2 \right).$$


*Author contributions.* The coupled CICE-PDAF model was developed by NB with support from LN and NW. NW further developed CICE-PDAF for assimilation of sea ice observations. Observations were processed for assimilation and evaluation by NW, DS and AR. NW performed the simulations and analysis under the supervision of DF, DS, PL, RB and NB. NB and DS provided additional technical support.
The manuscript was composed by NW with contributions and feedback from all authors.

*Competing interests.* The authors declare that they have no conflict of interest.

*Acknowledgements.* We would like to thank Maria Broadridge for her technical support during this work. CICE-PDAF simulations were performed on the ARCHER2 UK national supercomputing service.



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
