# Peer review of "The effects of assimilating a sub-grid scale sea ice thickness distribution in a new Arctic sea ice data assimilation system"

_EGUsphere, 2022_

## Author Comment (AC1)

Response to referee comments - Referee 1

Referee comments are shown in black, our response in blue, and changes to the manuscript in red.

Summary

A standalone sea ice model (CICE5.1.2) is used to investigate the impact of incorporating a sub-grid scale sea ice thickness distribution by coupling to the LETKF using the latest version of PDAF. The source of the ice thickness data is from monthly means of CryoSat-2. Multiple experiments are performed consisting of a control run (no assimilation), assimilation of ice concentration only (NASA Bootstrap), assimilation of ice concentration and mean ice thickness, and assimilation of ice concentration, mean ice thickness, and a monthly sea ice thickness distribution. Experiments with 100 ensemble members were performed in which ensemble spread was generated by perturbing the NCEP-2 atmospheric forcing. They find that a forgetting factor of 0.995, amplification factor of 1.5 and localization radius of 100km worked best in these studies. The authors state that this is the first time that a sub-grid scale thickness distribution product has been assimilated. The authors find that the experiment assimilating concentration, mean ice thickness and sub-grid scale thickness distribution performed best in the four thinnest sea ice categories. Comparisons were made against unassimilated CryoSat-2 observations.

I find this to be a well written paper with a thorough description of the techniques and analysis methods used. The graphics and tables are well laid out. I find that this research will be valuable to the community. I recommend publication with minor revisions noted below. General and specific comments are below.

Response: We would like to thank the reviewer for their time spent reviewing the paper and their helpful feedback.

General Comments:

Use CryoSat-2 (not Cryosat-2) throughout the paper.

In the section with lines 55-60; please add these additional references for model forecast systems assimilating sea ice concentration:

Smith GC, Roy F, Rezka M, Surcel Colan D, He Z, Deacu D, Bélanger J-M, Skachko S, Liu Y, Dupont F, Lemieux J-F, Beaudoin C, Tranchant B, Drévillon M, Garric G, Testut C-E, Lellouche J-M, Pellerin P, Ritchie H, Lu Y, Davidson F, Buehner M, Caya A, Lajoie M. 2014. Sea ice forecast verification in the Canadian Global Ice Ocean Prediction System. Q. J. R. Meteorol. Soc., https://doi.org/10.1002/qj.2555

Hebert, D. A., R. A. Allard, E. J. Metzger,P. G. Posey, R. H. Preller, A. J. Wallcraft,M. W. Phelps, and O. M. Smedstad(2015), Short-term sea ice forecasting:An assessment of ice concentrationand ice drift forecasts using the U.S.Navy's Arctic Cap Nowcast/ForecastSystem, J. Geophys. Res. Oceans, 120,8327–8345, doi:10.1002/2015JC011283.

Suggested change in manuscript: thank you, added additional references to model forecast systems

Papers by Massonnet et al. (2011) and Smith et al. (2022) examined the impact of a 15-category ice thickness distribution on seasonal and climate modeling studies. Please speculate on the potential impact of increasing the number of ice categories (ignoring the additional computational cost) in your

study.

Massonnet, F., Fichefet, T., Goosse, H., Vancoppenolle, M., Mathiot, P., and K.nig Beatty, C. (2011). On the influence of model physics on simulations of Arctic and Antarctic sea ice. The Cryosphere, 5(3), 687–699. https://doi.org/10.5194/tc-5-687-2011

Smith, M. M., Holland, M. M., Petty, A. A., Light, B., and Bailey, D. A. (2022). Effects of increasing the category resolution of the sea ice thickness distribution in a coupled climate model on Arctic and Antarctic sea ice mean state. Journal of Geophysical Research: Oceans, 127, e2022JC019044. https://doi.org/10.1029/

If we were to increase the number of ice thickness categories in the model, it would also require the reprocessing of the thickness distribution observations, and we would like there to be a sufficient number of data points in each category for the data to be meaningful. For 15 categories this would be much more difficult and there would be little data in many of the categories. We considered a separate experiment in this study with how the assimilation effects are changed by the number of ice thickness categories. A larger number of categories can represent the ice thickness distribution more realistically, however the thickness distribution observations provided are for the chosen five categories (default in CICE). As you mention the computational cost for our large ensemble run would also increase considerably. Given these limitations we have decided to stick with 5 categories. An increase of categories would be beneficial if the observations can support this. We will add a justification in the manuscript and a discussion of the possibility of increasing categories within the discussion section.

In lines 365-370 you state that using a forgetting factor of 0.995 (Fig. 1) does not lead to any model crashing. What is the cause of spikes seen in January – May, and Oct-Dec, evident in all runs except for the control?

These spikes are caused by the ice thickness assimilation which occurs between January and April and October to December, the assimilation happens at one timestep in each month, and the updates can be large in the central ice pack where the observation errors are lower, so this can cause jumps in the thickness and volume estimates. These also occur in concentration/area/extent on the first day that assimilation of concentration takes place.

In lines 430-432 you state: "for the assimilation runs that the decrease in concentration in late August in the Fram Strait leads to a remarkable increase in the sea ice thickness at the same time in these runs." I do not see any "remarkable increase". Please clarify, reword, or delete this sentence.

Suggested change in manuscript: Sentence deleted, thank you.

Lines 537-538: I disagree that all runs with assimilation of sea ice concentration showed very similar results. I agree they are similar to Bootstrap for any given year, but not amongst themselves. Please reword this section or provide additional details to me on what I seem to be missing.

Suggested change in manuscript: Clarified to mean that they are similar within the given year

Specific Comments:

Line 26: "rise at roughly twice this" amount.

Suggested change in manuscript: added quotation marks

Line 63: A comparison of fourteen ocean-sea ice reanalyzes (provide reference)

Suggested change in manuscript: thank you, added reference

Line 95: Somewhere in this section, please provide the horizontal resolution of the CICE model used in this study.

Suggested change in manuscript: added reference to horizontal resolution where grid cell sizes are discussed.

Line 163: (Gaspari and Cohn, 1999) do not appear in references. Please add.

Suggested change in manuscript: added to references, thank you.

Line 170: Is a reference missing where I see a "?" ?

Suggested change in manuscript: reference to previous section was missing, corrected.

Line 238: Reword phrase "Grid cells In CICE-PDAF we use..." awkward

Suggested change in manuscript: removed words "Grid cells" which don't belong in this sentence.

Line 333: "and CS2 thickness observations are assimilated monthly". Please clarify as there are not CS2 observations available for May – September.

Suggested change in manuscript: clarified that thickness observations are not available between May and September.

Figure 2: label on top and bottom for "c" and "h" I assume should be "assim_conc_hi_loc100?

As localisation of 100 km is used for further experiments in the paper, I only use assim_conc_hi for 100 km loc experiment here and for the remainder of the paper to

Figure 2 caption should be "Columns show CryoSat-2 and 4 CICE-PDAF runs..."

Suggested change in manuscript: Thanks, corrected.

Figure 3: Legend should be "assim_conc_hi_amp2" The "amp2" is missing.

Suggested change in manuscript: Figure has been corrected - see further below.

Line 450: I do not see a gold line in the legend.

Suggested change in manuscript: Should refer to black line, corrected.

Figure 6: With the exception of the control run (green), the 3 other experiments are difficult to see except for the assim_conc_hi_4hd. They must be very similar. Can assim_conc_hi_4hd be drawn first? Maybe assim_conc and assim_conc_hi will be easier to see.

Suggested change in manuscript: Will draw assim_conc_hi_4hd and also specify that other experiments are difficult to see as they are similar.

Line 468: Table 2 shows value of "0.62". Which is correct?

Suggested change in manuscript: Should be 0.63 in table, adjusted.

Lines 507-508: The assimilation of only concentration does not show an increase versus the control in the first year. Please modify sentence.

Suggested change in manuscript: modified sentence to say that first year ice volume is not increased by assimilation of only concentration in first year.

Line 511: I do not see a gold line in Fig 10. Please clarify.

Suggested change in manuscript: Should refer to black line, corrected.

Line 795: Hollinger et al. reference not cited.

Suggested change in manuscript: Added to reference list, thanks.

Line 878: Zhang and Krishnamurti (1999) not cited.

Suggested change in manuscript: Added to reference list, thanks.

---

## Author Comment (AC2)

Response to referee comments - Referee 2

Referee comments are shown in black, our response in blue, and changes to the manuscript in red.

The paper describes assimilation of monthly mean sea ice thickness, monthly mean sea ice thickness distribution from CryoSat-2 and sea ice concentration from passive microwave observations in CICE Arctic sea ice model. The best result was achieved when all three data sources are assimilated. For the first time, a sub-grid scale sea ice thickness distribution was assimilated. This led to a significant improvement in estimation of the thickness of the thickest ice category. The paper can be published once the following comments are addressed.

Response: We would like to thank the reviewer for their time spent reviewing the paper and their helpful feedback.

General Comment:

My major concern is that the only data source used for model verification was CryoSat-2 ice thicknesses that were not assimilated. In general, the fact that the modelled ice thickness becomes closer to CryoSat-2 observations indicate that the assimilation of CryoSat-2 ice thicknesses has been performed correctly. However, in order to further assess if the data assimilation brings the model closer to the reality, it is important to do a comparison against additional sea ice thickness observations. I suggest that the authors conduct verification of the model results against the available independent sea ice thickness observations including (1) NASA's IceBridge, (2) Beaufort Gyre Exploration Project upper-looking sonars (ULS), and (3) Airborne EM observations ("EM-bird"). I think that such additional verification analysis will substantially increase the quality of the paper.

Response: The difficulty of using these data sets is that they represent different space (and time) scales from the model, with its 40km by 40km grid cell size. In theory such comparison can be made useful if we add so-called representation errors to the measurement errors. These representation errors account for the scale mismatch and are typically much larger than the measurement errors. The difficulty, however, is in determining these representation errors. As discussed in Van Leeuwen (2015) there are essentially two ways to solve this problem. The first relies on having many observations evenly distributed in a grid box, and the second relies on running models at the spatial resolution of the observations. The first is not available, and the second is not feasible in our study. Alternatively, statistical methods based on data assimilation can be used, but those rely heavily on unbiasedness of the model, which is questionable. However, we agree that a comparison of the thickness with an independent data set for data assimilation can be important so we have conducted a comparison with Operation IceBridge, please see the figures of scatter plots of CICE-PDAF runs and CryoSat-2 against IceBridge and table showing RMSE comparisons below. Operation IceBridge is much more suited for comparisons of ice thickness for evaluation than the other two datasets proposed, because it has a much larger amount of data available concurrently with the time period of our experiment. The EM-Bird campaigns took place for some days in March and April each year between 2012 and 2015, which is around the same time that Operation IceBridge campaigns take place, but with less data available than IceBridge for validation. The EM-Bird data also provides ice + snow thickness together, not seperately, which introduces additional uncertainty for the validation of ice thickness for this study (though the data is not necessarily less reliable than Operation IceBridge, and very reliable over level ice (Haas et al., 2009) ). The BGEP ULS moorings provide ice draft which must be converted to ice thickness, either using a simple multiplication as in Rothrock et al (2003), or more accurately, by using snow depth – because the snow depth is very important in converting ice draft to ice thickness especially outside of summer. Uncertainty in snow density will also affect the conversion of draft to

thickness and comparisons. Of course the way CryoSat-2 uses the climatology to estimate snow depth is far from ideal. In Fiedler et al (2022), a paper which looked at assimilation of along-track sea ice thicknesses, found that the converted ice thicknesses from BGEP ULS moorings did not necessarily compare well to the CryoSat-2 observations of ice thickness, which meant that the assimilation of the CryoSat-2 ice thickness observations worsened the validation with the BGEP ULS data in comparison to the control. They also bring up the uncertainty in the conversion from sea ice draft to snow caused by ignoring the snow depth issue. From the analysis with Operation IceBridge, we can also conclude that comparing large area averages with point averages is erroneous and not necessarily useful for the other data sets where dealing with the snow depth could introduce additional errors. In terms of the comparison with Operation IceBridge, the assimilation of thickness (assim_conc_hi) and thickness distribution (assim_conc_hi_4hd) provide some improvement over the control towards Operation IceBridge, with an RMSE that is also similar to that between the CryoSat-2 data and the Operation IceBridge data (where we reiterate that it is unclear what the means exactly as the uncertainties in the IceBridge data related to the scale mismatch are unknown). Assimilating thickness distribution alongside the mean thickness did not improve the thickness RMSE much at all (only 0.01 m) toward the IceBridge thickness data. Assimilating the Bootstrap concentration alone (assim_conc) in our model significantly worsened the estimates of ice thickness against Operation IceBridge, as we already saw when we compared it against the CryoSat-2 estimates.

**Table 1** RMSE of sea ice thickness (m) for the four CICE-PDAF experiments against Operation IceBridge. The control experiment features no assimilation, assim_conc features assimilation of ice concentration, assim_conc_hi assimilation of concentration and thickness, and assim_conc_hi_4hd features assimilation of concentration, thickness and thickness distribution. This compares daily sea ice thickness over grid cells and on days where Operation IceBridge data is available (For each year, around 15 days in March and April).

| Experiment | Ice Thickness RMSE (m) |
|---|---|
| **control** | 0.64 |
| **assim_conc** | 0.92 |
| **assim_conc_hi** | 0.58 |
| **assim_conc_4hd** | 0.57 |
| **Operation IceBridge** | 0.53 |

Suggested change in manuscript: We will add a section explaining Operation IceBridge data and an evaluation against the CICE-PDAF experiments we have conducted using the figures and tables included in this response, and include a discussion of this comparison within the discussion section - including the evaluation of ice thickness in other studies.

Specific Minor Comments:

I noticed some minor inaccuracies throughout the paper:

Line 16 and throughout the paper. "Canadian Archipelago" à "Canadian Arctic Archipelago". Abbreviation "CAA" can be also used.

Suggested change in manuscript: Thank you, changed all references to CAA
Fig. 1 In Y-axis title please add ")". X-axis title should be "Month"?
Suggested change in manuscript: Thank you, adjusted plot

[Figure]

Figure 1: Scatter plots of mean sea ice thickness estimates in CICE-PDAF runs against Operation IceBridge for March and April 2012-2015

[Figure]

Figure 2: Scatter plots of mean sea ice thickness estimates in CryoSat-2 against Operation IceBridge for March and April 2012-2015

Line 58. Term "Optimal Interpolation" has been already introduced above.
Suggested change in manuscript: Yes, changed to introduce OI term
Line 170. Please define "(?)".
Suggested change in manuscript: reference to previous section was missing, corrected.

Line 190-191. Words "using", "use", "using" are part of the same sentence. Please rephrase.
Suggested change in manuscript: Rephrased sentence, thank you.

Line 214. Word "key" could be removed.
Suggested change in manuscript: Removed, thank you.

Line 217. "MYI", "FYI" were not defined.
Suggested change in manuscript: Defined both terms in first use of manuscript.

Line 218. "radiances" -¿ "brightness temperatures".
Suggested change in manuscript: Adjusted, thank you.

Line 226. "team" -¿ "Team".
Suggested change in manuscript: Adjusted, thank you.

Line 279. Please rephrase "Ice thinner than 0.5 is not assimilated. . ." − > "Ice thicknesses lower than 0.5 are not assimilated. . ."
Suggested change in manuscript: Rephrased, thank you.

Lines 399-401. "because" is used twice in the sentence.
Suggested change in manuscript: Rephrased, thank you.

Legend font on Fig. 5 is very small.
Suggested change in manuscript: Plot modified to increase legend.

Table 3. Add dimension to RMSE.
Suggested change in manuscript: Added, thank you.

Figure 8, y-axis title. Add dimension.
Suggested change in manuscript: Added, thank you.

Line 513. Should it be Fig. 13 instead of Fig. 12?
Suggested change in manuscript: Yes, corrected.

References
Fiedler, E.K., Martin, M.J., Blockley, E., Mignac, D., Fournier, N., Ridout, A., Shepherd, A. and Tilling, R., 2022. Assimilation of sea ice thickness derived from CryoSat-2 along-track freeboard measurements into the Met Office's Forecast Ocean Assimilation Model (FOAM). The Cryosphere, 16(1), pp.61-85.

Haas, C., Lobach, J., Hendricks, S., Rabenstein, L. and Pfaffling, A., 2009. Helicopter-borne measurements of sea ice thickness, using a small and lightweight, digital EM system. Journal of Applied Geophysics, 67(3), pp.234-241.

Rothrock, D.A., Zhang, J. and Yu, Y., 2003. The arctic ice thickness anomaly of the 1990s: A consistent view from observations and models. Journal of Geophysical Research: Oceans, 108(C3).

Van Leeuwen, P.J. (2015) Representation errors in Data Assimilation, Q. J. R. Meteorol. Soc., 2014, DOI: 10.1002/qj.2464